# A Discrete Variational Recurrent Topic Model without the Reparametrization Trick

**Mehdi Rezaee**
Department of Computer Science
University of Maryland Baltimore County
Baltimore, MD 21250 USA
rezaee1@umbc.edu

**Francis Ferraro**
Department of Computer Science
University of Maryland Baltimore County
Baltimore, MD 21250 USA
ferraro@umbc.edu

## Abstract

We show how to learn a neural topic model with discrete random variables—one that explicitly models each word's assigned topic—using neural variational inference that does not rely on stochastic backpropagation to handle the discrete variables. The model we utilize combines the expressive power of neural methods for representing sequences of text with the topic model's ability to capture global, thematic coherence. Using neural variational inference, we show improved perplexity and document understanding across multiple corpora. We examine the effect of prior parameters both on the model and variational parameters, and demonstrate how our approach can compete and surpass a popular topic model implementation on an automatic measure of topic quality.

## 1 Introduction

With the successes of deep learning models, neural variational inference (NVI) [27]—also called variational autoencoders (VAE)—has emerged as an important tool for neural-based, probabilistic modeling [19, 40, 37]. NVI is relatively straight-forward when dealing with continuous random variables, but necessitates more complicated approaches for discrete random variables.

The above can pose problems when we use discrete variables to model data, such as capturing both syntactic and semantic/thematic word dynamics in natural language processing (NLP). Short-term memory architectures have enabled Recurrent Neural Networks (RNNs) to capture local, syntactically-driven lexical dependencies, but they can still struggle to capture longer-range, thematic dependencies [44, 29, 5, 20]. Topic modeling, with its ability to effectively cluster words into thematically-similar groups, has a rich history in NLP and semantics-oriented applications [4, 3, 50, 6, 46, 35, i.a.]. However, they can struggle to capture shorter-range dependencies among the words in a document [8]. This suggests these problems are naturally complementary [42, 53, 45, 41, 27, 15, 8, 48, 51, 22].

NVI has allowed the above recurrent topic modeling approaches to be studied, but with two primary modifications: the discrete variables can be reparametrized and then sampled, or each word's topic assignment can be analytically marginalized out, *prior* to performing any learning. However, previous work has shown that topic models that preserve explicit topics yield higher quality topics than similar models that do not [25], and recent work has shown that topic models that have relatively consistent word-level topic assignments are preferred by end-users [24]. Together, these suggest that there are benefits to preserving these assignments that are absent from standard RNN-based language models. Specifically, preservation of word-level topics within a recurrent neural language model may both improve language prediction as well as yield higher quality topics.

To illustrate this idea of thematic vs. syntactic importance, consider the sentence "*She received bachelor's and master's degrees in electrical engineering from Anytown University.*" While a topic

model can easily learn to group these "education" words together, an RNN is designed explicitly to capture the sequential dependencies, and thereby predict coordination (*"and"*), metaphorical (*"in"*) and transactional (*"from"*) concepts given the previous *thematically*-driven tokens.

In this paper we reconsider core modeling decisions made by a previous recurrent topic model [8], and demonstrate how the discrete topic assignments can be maintained in both learning and inference without resorting to reparametrizing them. In this reconsideration, we present a simple yet efficient mechanism to learn the dynamics between thematic and non-thematic (e.g., syntactic) words. We also argue the design of the model's priors still provides a key tool for language understanding, even when using neural methods. The main contributions of this work are:

1. We provide a recurrent topic model and NVI algorithm that (a) explicitly considers the surface dynamics of modeling with thematic and non-thematic words and (b) maintains word-level, discrete topic assignments without relying on reparametrizing or otherwise approximating them.

2. We analyze this model, both theoretically and empirically, to understand the benefits the above modeling decisions yield. This yields a deeper understanding of certain limiting behavior of this model and inference algorithm, especially as it relates to past efforts.

3. We show that careful consideration of priors and the probabilistic model still matter when using neural methods, as they can provide greater control over the learned values/distributions. Specifically, we find that a Dirichlet prior for a document's topic proportions provides fine-grained control over the statistical structure of the learned variational parameters vs. using a Gaussian distribution.

Our code, scripts, and models are available at `https://github.com/mmrezaee/VRTM`.

## 2   Background

Latent Dirichlet Allocation [4, LDA] defines an admixture model over the words in documents. Each word $w_{d,t}$ in a document $d$ is stochastically drawn from one of $K$ topics—discrete distributions over $\mathcal{V}$ vocabulary words. We use the discrete variable $z_{d,t}$ to represent the topic that the $t$th word is generated from. We formally define the well-known generative story as drawing document-topic proportions $\theta_d \sim \text{Dir}(\alpha)$, then drawing individual topic-word assignments $z_{d,t} \sim \text{Cat}(\theta_d)$, and finally generating each word $w_{d,t} \sim \text{Cat}(\beta_{z_t})$, based on the topics ($\beta_{k,v}$ is the conditional probability of word $v$ given topic $k$).

Two common ways of learning an LDA model are either through Monte Carlo sampling techniques, that iteratively sample states for the latent random variables in the model, or variational/EM-based methods, which minimize the distribution distance between the posterior $p(\theta, z|w)$ and an approximation $q(\theta, z; \gamma, \phi)$ to that posterior that is controlled by learnable parameters $\gamma$ and $\phi$. Commonly, this means minimizing the negative KL divergence, $-\text{KL}(q(\theta, z; \gamma, \phi) \| p(\theta, z|w))$ with respect to $\gamma$ and $\phi$. We focus on variational inference as it cleanly allows neural components to be used.

Supported by work that has studied VAEs for a variety of distributions [12, 33], the use of deep learning models in the VAE framework for topic modeling has received recent attention [42, 53, 45, 41, 27, 15]. This complements other work that focused on extending the concept of topic modeling using undirected graphical models [43, 17, 21, 31].

Traditionally, stop words have been a nuisance when trying to learn topic models. Many existing algorithms ignore stop-words in initial pre-processing steps, though there have been efforts that try to alleviate this problem. Wallach et al. [47] suggested using an asymmetric Dirichlet prior distribution to include stop-words in some isolated topics. Eisenstein et al. [9] introduced a constant background factor derived from log-frequency of words to avoid the need for latent switching, and Paul [36] studied structured residual models that took into consideration the use of stop words. These approaches do not address syntactic language modeling concerns.

A recurrent neural network (RNN) can address those concerns. A basic RNN cell aims to predict each word $w_t$ given previously seen words $w_{<t} = \{w_0, w_1, \ldots, w_{t-1}\}$. Words are represented with embedding vectors and the model iteratively computes a new representation $h_t = f(w_{<t}, h_{t-1})$, where $f$ is generally a non-linear function.

There have been a number of attempts at leveraging the combination of RNNs and topic models to capture local and global semantic dependencies [8, 48, 51, 22, 39]). Dieng et al. [8] passed the

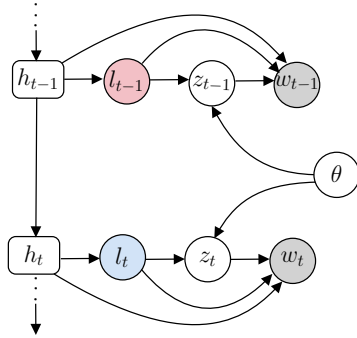

1. Draw a document topic vector $\theta \sim \text{Dir}(\alpha)$.
2. For each of the $T$ observed words ($t = 1 \ldots T$):
   - Compute the recurrent representation $h_t = f(x_t, h_{t-1})$, where $x_t = w_{t-1}$.
   - Draw $l_t \sim \text{Bern}(\rho)$, indicating whether word $t$ is thematically relevant.
   - Draw a topic $z_t \sim p(z_t | l_t, \theta)$.
   - Draw the word $w_t \sim p(w_t | z_t, l_t; h_t, \beta)$ using eq. 2.

(a) Graphical structure underlying the VRTM model. The shaded gray circles depict observed words. red nodes denotes non-thematic indicators ($l_t = 0$) and blue nodes show a thematic indicators ($l_t = 1$).

(b) The generative story of the model we study. The story largely follows Dieng et al. [8], though we redefine the prior on $\theta$ and the decoder $p(w_t | z_t, l_t; h_t, \beta)$. Beyond these, the core differences lie in how we approach learning the model.

Figure 1: An unrolled graphical model (Fig. 1a) for the corresponding generative story (Fig. 1b). Tokens are fed to a recurrent neural network (RNN) cell, which computes a word-specific representation $h_t$. With this representation, the thematic word indicator $l_t$, and inferred topic proportions $\theta$ shared across the document, we assign the topics $z_t$, which allows us to generate each word $w_t$. Based on our decoding model (eq. 2), the thematic indicators $l_t$ topic assignments help trade-off between the recurrent representations and the (stochastically) assigned topics $z_t$.

topics to the output of RNN cells through an additive procedure while Lau et al. [22] proposed using convolutional filters to generate document-level features and then exploited the obtained topic vectors in a gating unit. Wang et al. [48] provided an analysis of incorporating topic information inside a Long Short-Term Memory (LSTM) cell, followed by a topic diversity regularizer to make a complete compositional neural language model. Nallapati et al. [34] introduced sentence-level topics as a way to better guide text generation: from a document's topic proportions, a topic for a sentence would be selected and then used to generate the words in the sentence. The generated topics are shared among the words in a sentence and conditioned on the assigned topic. Li et al. [23] followed the same strategy, but modeled a document as a sequence of sentences, such that recurrent attention windows capture dependencies between successive sentences. Wang et al. [49] recently proposed using a normal distribution for document-level topics and a Gaussian mixture model (GMM) for word topics assignments. While they incorporate both the information from the previous words and their topics to generate a sentence, their model is based on a mixture of experts, with word-level topics neglected. To cover both short and long sentences, Gupta et al. [14] introduced an architecture that combines the LSTM hidden layers and the pre-trained word embeddings with a neural auto-regressive topic model. One difference between our model and theirs is that their focus is on predicting the words within a context, while our aim is to generate words given a particular topic without marginalizing.

## 3 Method

We first formalize the problem and propose our variationally-learned recurrent neural topic model (**VRTM**) that accounts for each word's generating topic. We discuss how we learn this model and the theoretical benefits of our approach. Subsequently we demonstrate the empirical benefits of our approach (Sec. 4).

### 3.1 Generative Model

Fig. 1 provides both an unrolled graphical model and the full generative story. We define each document as a sequence of $T$ words $\mathbf{w}_d = \{w_{d,t}\}_{t=1}^T$. For simplicity we omit the document index $d$ when possible. Each word has a corresponding generating topic $z_t$, drawn from the document's topic proportion distribution $\theta$. As in standard LDA, we assume there are $K$ topics, each represented as a $\mathcal{V}$-dimensional vector $\beta_k$ where $\beta_{k,v}$ is the probability of word $v$ given topic $k$. However, we sequentially model each word via an RNN. The RNN computes $h_t$ for each token, where $h_t$ is designed to consider the previous $t - 1$ words observed in sequence. During training, we define a

sequence of observed semantic indicators (one per word) $\mathbf{l} = \{l_t\}_{t=1}^T$, where $l_t = 1$ if that token is thematic (and 0 if not); a sequence of latent topic assignments $\mathbf{z} = \{z_t\}_{t=1}^T$; and a sequence of computed RNN states $\mathbf{h} = \{h_t\}_{t=1}^T$. We draw each word's topic from an assignment distribution $p(z_t|l_t, \theta)$, and generate each word from a decoding/generating distribution $p(w_t|z_t, l_t; h_t, \beta)$.

The joint model can be decomposed as

$$p(\mathbf{w}, \mathbf{l}, \mathbf{z}, \theta; \beta, \mathbf{h}) = p(\theta; \alpha) \prod_{t=1}^T p(w_t|z_t, l_t; h_t, \beta) p(z_t|l_t, \theta) p(l_t; h_t), \tag{1}$$

where $\alpha$ is a vector of positive parameters governing what topics are likely in a document ($\theta$). The corresponding graphical model is shown in Fig. 1a. Our decoder is

$$p(w_t = v|z_t = k; h_t, l_t, \beta) \propto \exp\left(p_v^\top h_t + l_t \beta_{k,v}\right), \tag{2}$$

where $p_v$ stands for the learned projection vector from the hidden space to output. When the model faces a non-thematic word it just uses the output of the RNN and otherwise it uses a mixture of LDA and RNN predictions. As will be discussed in Sec. 3.2.2, separating $h_t$ and $z_t = k$ in this way allows us to marginalize out $z_t$ during inference without reparametrizing it.

A similar type of decoder has been applied by both Dieng et al. [8] and Wen and Luong [51] to combine the outputs of the RNN and LDA. However, as they marginalize out the topic assignments their decoders do not condition on the topic assignment $z_t$. Mathematically, their decoders compute $\beta_v^\top \theta$ instead of $\beta_{k,v}$, and compute $p(w_t = v|\theta; h_t, l_t, \beta) \propto \exp\left(p_v^\top h_t + l_t \beta_v^\intercal \theta\right)$. This new decoder structure helps us preserve the word-level topic information which is neglected in other models. This seemingly small change has out-sized empirical benefits (Tables 1b and 2).[1]

Note that we explicitly allow the model to trade off between thematic and non-thematic words. This is very much similar to Dieng et al. [8]. We assume during training these indicators are observable, though their occurrence is controlled via the output of a neural factor: $\rho = \sigma\left(g(h_t)\right)$ ($\sigma$ is the sigmoid function). Using the recurrent network's representations, we allow the model to learn the presence of thematic words to be learned via—a long recognized area of interest [11, 26, 52, 16].

For a $K$-topic VRTM, we formalize this intuition by defining $p(z_t = k|l_t, \theta) = \frac{1}{K}$, if $l_t = 0$ and $\theta_k$ otherwise. The rationale behind this assumption is that in our model the RNN is sufficient to draw non-thematic words and considering a particular topic for these words makes the model unnecessarily complicated. Maximizing the data likelihood $p(\mathbf{w}, \mathbf{l})$ in this setting is intractable due to the integral over $\theta$. Thus, we rely on amortized variational inference as an alternative approach. While classical approaches to variational inference have relied on statistical conjugacy and other approximations to make inference efficiently computable, the use of neural factors complicates such strategies. A notable difference between our model and previous methods is that we take care to account for each word and its assignment in both our model and variational approximation.

## 3.2 Neural Variational Inference

We use a mean field assumption to define the variational distribution $q(\theta, \mathbf{z}|\mathbf{w}, \mathbf{l})$ as

$$q(\theta, \mathbf{z}|\mathbf{w}, \mathbf{l}) = q(\theta|\mathbf{w}, \mathbf{l}; \gamma) \prod_{t=1}^T q(z_t|w_t, l_t; \phi_t), \tag{3}$$

where $q(\theta|\mathbf{w}, \mathbf{l}, \gamma)$ is a Dirichlet distribution parameterized by $\gamma$ and we define $q(z_t)$ similarly to $p(z_t)$: $q(z_t = k|w_t, l_t; \phi_t) = \frac{1}{K}$ if $l_t = 0$ and the neural parametrized output $\phi_t^k$ otherwise. In Sec. 3.2.1 we define $\gamma$ and $\phi_t$ in terms of the output of neural factors. These allow us to "encode" the input into our latent variables, and then "decode" them, or "reconstruct" what we observe.

Variational inference optimizes the evidence lower bound (ELBO). For a single document (subscript omitted for clarity), we write the ELBO as:

$$\mathcal{L} = \mathbb{E}_{q(\theta, \mathbf{z}|\mathbf{w}, \mathbf{l})} \left[ \sum_{t=1}^T \log p(w_t|z_t, l_t; h_t, \beta) + \log \frac{p(z_t|l_t, \theta)}{q(z_t|w_t, l_t)} + \log p(l_t; h_t) + \log \frac{p(\theta)}{q(\theta|\mathbf{w}, \mathbf{l})} \right]. \tag{4}$$

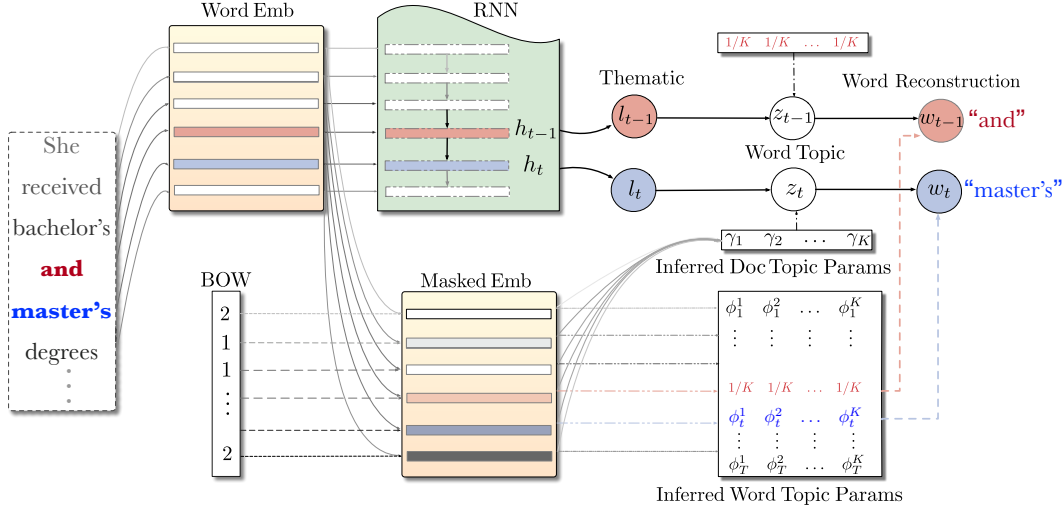

Figure 2: Our encoder and decoder architectures. Word embeddings, learned from scratch, are both provided to a left-to-right recurrent network and used to create bag-of-words masked embeddings (masking non-thematic words, i.e., $l_t = 0$, by 0). These masked embeddings are used to compute $\gamma$ and $\phi$—the variational parameters—while the recurrent component computes $h_t$ (which is used to compute the thematic predictor and reconstructed words, as shown in Fig. 1). For clarity, we have omitted the connection from $h_t$ to $w_t$ (these are shown in Fig. 1a).

This objective can be decomposed into separate loss functions and written as $\mathcal{L} = \mathcal{L}_w + \mathcal{L}_z + \mathcal{L}_\phi + \mathcal{L}_l + \mathcal{L}_\theta$, where, using $\langle \cdot \rangle = \mathbb{E}_{q(\theta, \mathbf{z}|\mathbf{w}, \mathbf{l})} [\cdot]$, we have $\mathcal{L}_w = \langle \sum_t \log p(w_t|z_t, l_t; h_t, \beta) \rangle$, $\mathcal{L}_z = \langle \sum_t \log p(z_t|l_t, \theta) \rangle$, $\mathcal{L}_\phi = \langle \sum_t \log q(z_t|w_t, l_t) \rangle$, $\mathcal{L}_l = \langle \sum_t \log p(l_t; h_t) \rangle$, and $\mathcal{L}_\theta = \langle \log \frac{p(\theta)}{q(\theta|\mathbf{w}, \mathbf{l})} \rangle$.

The algorithmic complexity, both time and space, will depend on the exact neural cells used. The forward pass complexity for each of the separate summands follows classic variational LDA complexity.

In Fig. 2 we provide a system overview of our approach, which can broadly be viewed as an encoder (the word embedding, bag-of-words, masked word embeddings, and RNN components) and a decoder (everything else). Within this structure, our aim is to maximize $\mathcal{L}$ to find the best variational and model parameters. To shed light onto what the ELBO is showing, we describe each term below.

### 3.2.1 Encoder

Our encoder structure maps the documents into variational parameters. Previous methods posit that while an RNN can adequately perform language modeling (LM) using word embeddings for all vocabulary words, a modified, bag-of-words-based embedding representation is an effective approach for capturing thematic associations [8, 48, 22]. We note however that more complex embedding methods can be studied in the future [38, 7, 2].

To avoid "representation degeneration" [13], care must be taken to preserve the implicit semantic relationships captured in word embeddings.[2] Let $\mathcal{V}$ be the overall vocabulary size. For RNN language modeling [LM], we represent each word $v$ in the vocabulary with an $E$-dimensional real-valued vector $e_v \in \mathbb{R}^E$. As a result of our model definition, the topic modeling [TM] component of VRTM effectively operates with a reduced vocabulary; let $\mathcal{V}'$ be the vocabulary size excluding non-thematic words ($\mathcal{V}' \leq \mathcal{V}$). Each document is represented by a $\mathcal{V}'$-dimensional integer vector $\mathbf{c} = \langle c_1, c_2, \cdots, c_{\mathcal{V}'} \rangle$, recording how often each thematic word appeared in the document.

In contrast to a leading approach [49], we use the entire $e_v$ vocabulary embeddings for LM and mask out the non-thematic words to compute the variational parameters for TM as $e_v^{\mathrm{TM}} = e_v \odot c_v^{\mathrm{TM}}$, where $\odot$ denotes element-wise multiplication and $c_v^{\mathrm{TM}} = c_v$ if $v$ is thematic (0 otherwise). We refer to $e_v^{\mathrm{TM}}$ as the masked embedding representation. This motivates the model to learn the word meaning

and importance of word counts jointly. The token representations are gathered into $e_w^{\text{TM}} \in \mathbb{R}^{T \times E}$, which is fed to a feedforward network with one hidden layer to infer word-topics $\phi$. Similarly, we use tensordot to produce the $\gamma$ parameters for each document defined as $\gamma = \text{softplus}(e_w^{\text{TM}} \otimes W_\gamma)$, where $W_\gamma \in \mathbb{R}^{T \times E \times K}$ is the weight tensor and the summation is over the common axes.

### 3.2.2 Decoder/Reconstruction Terms

The first term in the objective, $\mathcal{L}_w$, is the reconstruction part, which simplifies to

$$\mathcal{L}_w = \mathbb{E}_{q(\theta,\mathbf{z}|\mathbf{w},\mathbf{l})}\left[\sum_{t=1}^{T} \log p(w_t|z_t, l_t; h_t, \beta)\right] = \sum_{t=1}^{T}\left[\mathbb{E}_{q(z_t|w_t, l_t; \phi_t)} \log p(w_t|z_t, l_t; h_t, \beta)\right]. \quad (5)$$

For each word $w_t$, if $l_t = 0$, then the word is not thematically relevant. Because of this, we just use $h_t$. Otherwise ($l_t = 1$) we let the model benefit from both the globally semantically coherent topics $\beta$ and the locally fluent recurrent states $h_t$. With our definition of $p(z_t|\theta)$, $\mathcal{L}_w$ simplifies to $\mathcal{L}_w = \sum_{t:l_t=1} \sum_{k=1}^{K} \phi_t^k \log p(w_t|z_t; h_t, \beta) + \sum_{t:l_t=0} \log p(w_t; h_t)$. Finally, note that there is an additive separation between $h_t$ and $l_t \beta_{z_t,v}$ in the word reconstruction model (eq. 2). The marginalization that occurs per-word can be computed as $p_v^{\mathsf{T}} h_t + l_t \langle \beta_{z_t,v}\rangle_{z_t} - \langle \log Z(z_t, l_t; h_t, \beta)\rangle_{z_t}$, where $Z(z_t, l_t; h_t, \beta)$ represents the per-word normalization factor. This means that the discrete $z$ can be marginalized out and do not need to be sampled or approximated.

Given the general encoding architecture used to compute $q(\theta)$, computing $\mathcal{L}_z = \mathbb{E}_{q(\theta,\mathbf{z}|\mathbf{w},\mathbf{l})}\left[\sum_{t=1}^{T} \log p(z_t|l_t, \theta)\right]$ requires computing two expectations. We can exactly compute the expectation over $z$, but we approximate the expectation over $\theta$ with $S$ Monte Carlo samples from the learned variational approximation, with $\theta^{(s)} \sim q(\theta)$. Imposing the basic assumptions from $p(z)$ and $q(z)$, we have $\mathcal{L}_z \approx \frac{1}{S} \sum_{s=1}^{S} \sum_{t:l_t=1} \sum_{k=1}^{K} \phi_t^k \log \theta_k^{(s)} - C \log K$, where $C$ is the number of the non-thematic words in the document and $\theta_k^{(j)}$ is the output of Monte Carlo sampling. Similarly, we have $\mathcal{L}_\phi = -\mathbb{E}_{q(\mathbf{z}|\mathbf{w},\mathbf{l};\phi)}\left[\log q(\mathbf{z}|\mathbf{w},\mathbf{l};\phi)\right] = -\sum_{t:l_t=1} \sum_{k=1}^{K} \phi_t^k \log \phi_t^k + C \log K$.

Modeling which words are thematic or not can be seen as a negative binary cross-entropy loss: $\mathcal{L}_l = \mathbb{E}_{q(\theta,\mathbf{z}|\mathbf{w},\mathbf{l})}\left[\log p(l_t; h_t)\right] = \sum_{t=1}^{T} \log p(l_t; h_t)$. This term is a classifier for $l_t$ that motivates the output of the RNN to discriminate between thematic and non-thematic words. Finally, as a KL-divergence between distributions of the same type, $\mathcal{L}_\theta$ can be calculated in closed form [18].

We conclude with a theorem, which shows VRTM is an extension of RNN under specific assumptions. This demonstrates the flexibility of our proposed model and demonstrates why, in addition to natural language processing considerations, the renewed accounting mechanisms for thematic vs. non-thematic words in both the model *and* inference is a key idea and contribution. For the proof, please see the appendix. The central idea is to show that all the terms related to TM either vanish or are constants; it follows intuitively from the ELBO and eq. 2.

**Theorem 1.** *If in the generative story $\rho = 0$, and we have a uniform distribution for the prior and variational topic posterior approximation $\left(p(\theta) = q(\theta|\mathbf{w},\mathbf{l}) = 1/K\right)$, then VRTM reduces to a Recurrent Neural Network to just reconstruct the words given previous words.*

## 4 Experimental Results

**Datasets** We test the performance of our algorithm on the APNEWS, IMDB and BNC datasets that are publicly available.[3] Roughly, there are between 7.7k and 9.8k vocab words in each corpus, with between 15M and 20M training tokens each; Table A1 in the appendix details the statistics of these datasets. APNEWS contains 54k newswire articles, IMDB contains 100k movie reviews, and BNC contains 17k assorted texts, such as journals, books excerpts, and newswire. These are the same datasets including the train, validation and test splits, as used by prior work, where additional details can be found [48]. We use the publicly provided tokenization and following past work we lowercase all text and map infrequent words (those in the bottom $0.01\%$ of frequency) to a special <unk> token. Following previous work [8], to avoid overfitting on the BNC dataset we grouped 10 documents in the training set into a new pseudo-document.

**Setup** Following Dieng et al. [8], we find that, for basic language modeling and core topic modeling, identifying which words are/are not stopwords is a sufficient indicator of thematic relevance. We define $l_t = 0$ if $w_t$ is within a standard English stopword list, and 1 otherwise.[4] We use the softplus function as the activation function and batch normalization is employed to normalize the inputs of variational parameters. We use a single-layer recurrent cell; while some may be surprised at the relative straightforwardness of this choice, we argue that this choice is a benefit. Our models are intentionally trained using *only* the provided text in the corpora so as to remain comparable with previous work and limit confounding factors; we are not using pre-trained models. While transformer methods are powerful, there are good reasons to research non-transformer approaches, e.g., the data & computational requirements of transformers. We demonstrate the benefits of separating the "semantic" vs. "syntactic" aspects of LM, and could provide the blueprint for future work into transformer-based TM. For additional implementation details, please see appendix C.

**Language Modeling Baselines** In our experiments, we compare against the following methods:

- **basic-LSTM**: A single-layer LSTM with the same number of units like our method implementation but without any topic modeling, i.e. $l_t = 0$ for all tokens.
- **LDA+LSTM** [22]: Topics, from a trained LDA model, are extracted for each word and concatenated with the output of an LSTM to predict next words.
- **Topic-RNN** [8]: This work is the most similar to ours. They use the similar decoder strategy (Eq. 2) but marginalize topic assignments prior to learning.
- **TDLM** [22]: A convolutional filter is applied over the word embeddings to capture long range dependencies in a document and the extracted features are fed to the LSTM gates in the reconstruction phase.
- **TCNLM** [48]: A Gaussian vector defines document topics, then a set of expert networks are defined inside an LSTM cell to cover the language model. The distance between topics is maximized as a regularizer to have diverse topics.
- **TGVAE** [49]: The structure is like TCNLM, but with a Gaussian mixture model to define each sentence as a mixture of topics. The model inference is done by using $K$ householder flows to increase the flexibility of variational distributions.

### 4.1 Evaluations and Analysis

**Perplexity** For an RNN, we used a single-layer LSTM with 600 units in the hidden layer, set the size of embedding to be 400, and had a fixed/maximum sequence length of 45. We also present experiments demonstrating the performance characteristics of using basic RNN, GRU and LSTM cells in Table 1a. Note that although our embedding size is higher we do not use pretrained word embeddings, but instead learn them from scratch via end-to-end training.[5] The perplexity values of the baselines and our VRTM across our three heldout evaluation sets are shown in Table 1b. We note that TGVAE [49] also compared against, and outperformed, all the baselines in this work. In surpassing TGVAE, we are also surpassing them. We find that our proposed method achieves lower perplexity than LSTM. This is consistent with Theorem 1. We observe that increasing the number of topics yields better overall perplexity scores. Moreover, as we see VRTM outperforms other baselines across all the benchmark datasets.

**Topic Switch Percent** Automatically evaluating the quality of learned topics is notoriously difficult. While "coherence" [30] has been a popular automated metric, it can have peculiar failure points especially regarding very common words [36]. To counter this, Lund et al. [24] recently introduced switch percent (SwitchP). SwitchP makes the very intuitive yet simple assumption that "good" topics will exhibit a type of inertia: one would not expect adjacent words to use many different topics. It aims to show the consistency of the adjacent word-level topics by computing the number of times the same topic was used between adjacent words: $(T_d - 1)^{-1} \sum_{t=1}^{T_d - 1} \delta(z_t, z_{t+1})$, where $z_t = \arg\max_k \{\phi_t^1, \phi_t^2, \ldots, \phi_t^K\}$, and $T_d$ is the length of document after removing the stop-words.

| Model | APNEWS | | IMDB | |
|---|---|---|---|---|
| | **300** | **400** | **300** | **400** |
| RNN ($T = 10$) | 66.21 | 61.48 | 69.48 | 79.94 |
| RNN ($T = 30$) | 60.03 | 61.21 | 80.65 | 76.04 |
| RNN ($T = 50$) | 59.24 | 60.64 | 66.45 | 66.42 |
| GRU ($T = 10$) | 61.57 | 58.82 | 75.21 | 67.31 |
| GRU ($T = 30$) | 63.40 | 58.59 | 84.27 | 67.61 |
| GRU ($T = 50$) | 59.09 | 57.91 | 84.45 | 63.59 |
| LSTM ($T = 10$) | 55.19 | 54.31 | 60.14 | 59.82 |
| LSTM ($T = 30$) | 53.76 | 51.47 | 58.27 | 54.36 |
| LSTM ($T = 50$) | 51.35 | 47.78 | 57.70 | 51.08 |

| Methods | APNEWS | IMDB | BNC |
|---|---|---|---|
| basic-LSTM | 62.79 | 70.38 | 100.07 |
| LDA+LSTM ($T = 50$) | 57.05 | 69.58 | 96.42 |
| Topic-RNN ($T = 50$) | 56.77 | 68.74 | 94.66 |
| TDLM ($T = 50$) | 53.00 | 63.67 | 91.42 |
| TCNLM ($T = 50$) | 52.75 | 63.98 | 87.98 |
| TGVAE ($T = 10$) | $\leq$55.77 | $\leq$62.22 | $\leq$91.19 |
| TGVAE ($T = 30$) | $\leq$51.27 | $\leq$59.45 | $\leq$88.34 |
| TGVAE ($T = 50$) | $\leq$48.73 | $\leq$57.11 | $\leq$87.86 |
| VRTM-LSTM ($T = 10$) | $\leq$54.31 | $\leq$59.82 | $\leq$92.89 |
| VRTM-LSTM ($T = 30$) | $\leq$51.47 | $\leq$54.36 | $\leq$89.26 |
| VRTM-LSTM ($T = 50$) | $\leq$ **47.78** | $\leq$ **51.08** | $\leq$ **86.33** |

(a) Test perplexity for different RNN cells and embedding sizes (300 vs. 400). $T$ denotes the number of topics.

(b) Test perplexity, as reported in previous works.[6] $T$ denotes the number of topics. Consistent with Wang et al. [49] we report the maximum of three VRTM runs.

Table 1: Test set perplexity (lower is better) of VRTM demonstrates the effectiveness of our approach at learning a topic-based language model. In 1a we demonstrate the stability of VRTM using different recurrent cells. In 1b, we demonstrate our VRTM-LSTM model outperforms prior neural topic models. We do not use pretrained word embeddings.

| Topics | APNEWS | | IMDB | | BNC | |
|---|---|---|---|---|---|---|
| | LDA | VRTM | LDA | VRTM | LDA | VRTM |
| 5 | 0.26 | 0.59 | 0.24 | 0.52 | 0.24 | 0.51 |
| 10 | 0.18 | 0.43 | 0.14 | 0.35 | 0.15 | 0.40 |
| 15 | 0.14 | 0.33 | 0.12 | 0.31 | 0.13 | 0.35 |
| 30 | 0.10 | 0.31 | 0.09 | 0.28 | 0.10 | 0.23 |
| 50 | 0.08 | 0.20 | 0.07 | 0.26 | 0.07 | 0.20 |

| Topics | APNEWS | | IMDB | | BNC | |
|---|---|---|---|---|---|---|
| | LDA | VRTM | LDA | VRTM | LDA | VRTM |
| 5 | 1.61 | 0.91 | 1.60 | 0.96 | 1.61 | 1.26 |
| 10 | 2.29 | 1.65 | 2.29 | 1.56 | 2.30 | 1.76 |
| 15 | 2.70 | 1.69 | 2.71 | 2.10 | 2.71 | 1.77 |
| 30 | 3.39 | 2.23 | 3.39 | 2.54 | 3.39 | 2.22 |
| 50 | 3.90 | 2.63 | 3.90 | 2.74 | 3.90 | 2.64 |

(a) SwitchP (higher is better) for VRTM vs LDA VB [4] averaged across three runs. Comparisons are valid within a corpus and for the same number of topics.

(b) Average document-level topic $\theta$ entropy, across three runs, on the test sets. Lower entropy means a document prefers using fewer topics.

Table 2: We provide both SwitchP [24] results and entropy analysis of the model. These results support the idea that if topic models capture semantic dependencies, then they should capture the topics well, explain the topic assignment for each word, and provide an overall level of thematic consistency across the document (lower $\theta$ entropy).

SwitchP is bounded between 0 (more switching) and 1 (less switching), where higher is better. Despite this simplicity, SwitchP was shown to correlate significantly better with human judgments of "good" topics than coherence (e.g., a coefficient of determination of 0.91 for SwitchP vs 0.49 for coherence). Against a number of other potential methods, SwitchP was consistent in having high, positive correlation with human judgments. For this reason, we use SwitchP. However, like other evaluations that measure some aspect of meaning (e.g., BLEU attempting to measure meaning-preserving translations) comparisons can only be made in like-settings.

In Table 2a we compare SwitchP for our proposed algorithm against a variational Bayes implementation of LDA [4]. We compare to this classic method, which we call LDA VB, since both it and VRTM use variational inference: this lessens the impact of the learning algorithm, which can be substantial [25, 10], and allows us to focus on the models themselves.[7] We note that it is difficult to compare to many recent methods: our other baselines do not explicitly track each word's assigned topic, and evaluating them with SwitchP would necessitate non-trivial, core changes to those methods.[8] We push VRTM to associate select words with topics, by (stochastically) assigning topics. Because standard LDA VB does not handle stopwords well, we remove them for LDA VB, and to ensure a fair comparison, we mask all stopwords from VRTM. First, notice that LDA VB switches

| Dataset | #1 | #2 | #3 | #4 | #5 | #6 | #7 | #8 | #9 |
|---|---|---|---|---|---|---|---|---|---|
| **APNEWS** | dead | washington | soldiers | fund | police | state | car | republican | city |
| | killed | american | officers | million | death | voted | road | u.s. | residents |
| | hunting | california | army | bill | killed | voters | line | president | st. |
| | deaths | texas | weapons | finance | accusing | democrats | rail | campaign | downtown |
| | kill | residents | blaze | billion | trial | case | trail | candidates | visitors |

Table 3: Nine random topics extracted from a 50 topic VRTM learned on the APNEWS corpus. See Table A2 in the Appendix for topics from IMBD and BNC.

| Data | Generated Sentences |
|---|---|
| **APNEWS** | • a damaged car and body <unk> were taken to the county medical center from dinner with one driver. <br> • the house now has released the <unk> of $ 100,000 to former <unk> freedom u.s. postal service and $ <unk> to the federal government. <br> • another agency will investigate possible abuse of violations to the police facility . <br> • not even if it represents everyone under control . we are getting working with other items . <br> • the oklahoma supreme court also faces a maximum amount of money for a local counties. <br> • the money had been provided by local workers over how much <unk> was seen in the spring. <br> • he did n't offer any evidence and they say he was taken from a home in front of a vehicle. |
| **IMDB** | • the film is very funny and entertaining . while just not cool and all ; the worst one can be expected. <br> • if you must view this movie , then i 'd watch it again again and enjoy it .this movie surprised me . <br> • they definitely are living with characters and can be described as vast <unk> in their parts . <br> • while obviously not used as a good movie , compared to this ; in terms of performances . <br> • i love animated shorts , and once again this is the most moving show series i 've ever seen. <br> • i remember about half the movie as a movie . when it came out on dvd , it did so hard to be particularly silly . <br> • this flick was kind of convincing . john <unk> 's character better could n't be ok because he was famous as his sidekick. |
| **BNC** | • she drew into her eyes . she stared at me . molly thought of the young lady , there was lack of same feelings of herself. <br> • these conditions are needed for understanding better performance and ability and entire response . <br> • it was interesting to give decisions order if it does not depend on your society , measured for specific thoughts , sometimes at least . <br> • not a conservative leading male of his life under waste worth many a few months to conform with how it was available . <br> • their economics , the brothers began its $ <unk> wealth by potential shareholders to mixed them by tomorrow . <br> • should they happen in the north by his words , as it is a hero of the heart , then without demand . <br> • we will remember the same kind of the importance of information or involving agents what we found this time. |

Table 4: Seven randomly generated sentences from a VRTM model learned on the three corpora.

approximately twice as often. Second, as intuition may suggest, we see that with larger number of topics both methods' SwitchP decreases—though VRTM still switches less often. Third, we note sharp decreases in LDA VB in going from 5 to 10 topics, with a more gradual decline for VRTM. As Lund et al. [24] argue, these results demonstrate that our model and learning approach yield more "consistent" topic models; this suggests that our approach effectively summarizes thematic words consistently in a way that allows the overall document to be modeled.

**Inferred Document Topic Entropy** To better demonstrate characteristics of our learned model, we report the average document topics entropy with $\alpha = 0.5$ . Table 2b presents the results of our approach, along with LDA VB. These results show that the sparsity of relevant/likely topics are much more selective about which topics to use in a document. Together with the SwitchP and perplexity results, this suggests VRTM can provide consistent topic analyses.

**Topics & Sentences** To demonstrate the effectiveness of our proposed masked embedding representation, we report nine random topics in Table 3 (see Table A2 in the appendix for more examples), and seven randomly generated sentences in Table 4. During the training we observed that the topic-word distribution matrix ($\beta$) includes stop words at the beginning and then the probability of stop words given topics gradually decreases. This observation is consistent with our hypothesis that the stop-words do not belong to any specific topic. The reasons for this observation can be attributed to the uniform distributions defined for variational parameters, and to controlling the updates of $\beta$ matrix while facing the stop and non-stop words. A rigorous, quantitative examination of the generation capacity deserves its own study and is beyond the scope of this work. We provide these so readers may make their own qualitative assessments on the strengths and limitations of our methods.

# 5 CONCLUSION

We incorporated discrete variables into neural variational without analytically integrating them out or reparametrizing and running stochastic backpropagation on them. Applied to a recurrent, neural topic model, our approach maintains the discrete topic assignments, yielding a simple yet effective way to learn thematic vs. non-thematic (e.g., syntactic) word dynamics. Our approach outperforms previous approaches on language understanding and other topic modeling measures.

## Broader Impact

The model used in this paper is fundamentally an associative-based language model. While NVI does provide some degree of regularization, a significant component of the training criteria is still a cross-entropy loss. Further, this paper's model does not examine adjusting this cross-entropy component. As such, the text the model is trained on can influence the types of implicit biases that are transmitted to the learned syntactic component (the RNN/representations $h_t$), the learned thematic component (the topic matrix $\beta$ and topic modeling variables $\theta$ and $z_t$), and the tradeoff(s) between these two dynamics ($l_t$ and $\rho$). For comparability, this work used available datasets that have been previously published on. Based upon this work's goals, there was not an in-depth exploration into any biases within those datasets. Note however that the thematic vs. non-thematic aspect of this work provides a potential avenue for examining this. While we treated $l_t$ as a binary indicator, future work could involve a more nuanced, gradient view.

Direct interpretability of the individual components of the model is mixed. While the topic weights can clearly be inspected and analyzed directly, the same is not as easy for the RNN component. While lacking a direct way to inspect the *overall* decoding model, our approach does provide insight into the thematic component.

We view the model as capturing thematic vs. non-thematic dynamics, though in keeping with previous work, for evaluation we approximated this with non-stopword vs. stopword dynamics. Within topic modeling stop-word handling is generally considered *simply* a preprocessing problem (or obviated by neural networks), we believe that preprocessing is an important element of a downstream user's workflow that is not captured when preprocessing is treated as a stand-alone, perhaps boring step. We argue that future work can examine how different elements of a user's workflow, such as preprocessing, can be handled with our approach.

**Acknowledgements and Funding Disclosure**   We would like to thank members and affiliates of the UMBC CSEE Department, including Edward Raff, Cynthia Matuszek, Erfan Noury and Ahmad Mousavi. We would also like to thank the anonymous reviewers for their comments, questions, and suggestions. Some experiments were conducted on the UMBC HPCF. We'd also like to thank the reviewers for their comments and suggestions. This material is based in part upon work supported by the National Science Foundation under Grant No. IIS-1940931. This material is also based on research that is in part supported by the Air Force Research Laboratory (AFRL), DARPA, for the KAIROS program under agreement number FA8750-19-2-1003. The U.S.Government is authorized to reproduce and distribute reprints for Governmental purposes notwithstanding any copyright notation thereon. The views and conclusions contained herein are those of the authors and should not be interpreted as necessarily representing the official policies or endorsements, either express or implied, of the Air Force Research Laboratory (AFRL), DARPA, or the U.S. Government.

## Footnotes

[1] Since the model drives learning and inference, this model's similarities and adherence to both foundational [4] and recent neural topic models [8] are intentional. We argue, and show empirically, that this adherence yields language modeling performance and allows more exact marginalization during learning.

[2]"Representation degeneration" happens when after training the model, the word embeddings are not well separated in the embedding space and all of them are highly correlated [13].

[3]https://github.com/jhlau/topically-driven-language-model

[4]Unlike Asymmetric Latent Dirichlet Allocation (ALDA) [47], which assigns a specific topic for stop words, we assume that stop words *do not* belong to any specific topic.

[5]Perplexity degraded slightly with pretrained embeddings, e.g., with $T = 300$, VRTM-LSTM on APNEWS went from 51.35 (from scratch) to 52.31 (word2vec).

[7]Given its prominence in the topic modeling community, we initially examined Mallet. It uses Collapsed Gibbs Sampling, rather than variational inference, to learn the parameters—a large difference that confounds comparisons. Still, VRTM's SwitchP was often competitive with, and in some cases surpassed, Mallet's.

[8]For example, evaluating the effect of a recurrent component in a model like LDA+LSTM [22] is difficult, since the LDA and LSTM models are trained separately. While the LSTM model does include inferred document-topic proportions $\theta$, these values are concatenated to the computed LSTM hidden state. The LDA model is frozen, so there is no propagation back to the LDA model: the results would be the same as vanilla LDA.

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
