[Supplementary Material]

# Supplementary Material

## A    Proof of Theorm 1

Theorem 1 states: If in the generative story $\rho = 0$, and we have a uniform distribution for the prior and variational topic posterior approximation $\left( p(\theta) = q(\theta | \mathbf{w}, \mathbf{l}) = 1/K \right)$, then VRTM reduces to a Recurrent Neural Network to just reconstruct the words given previous words.

Following is the proof.

*Proof.* If $\rho = 0$ then for every token $t$, $l_t = 0$. Therefore, the reconstruction term simplifies to $\mathcal{L}_w = \sum_{t=1}^{T} \log p(w_t; h_t)$. We proceed to analyze the other remaining terms as following. The first terms on the right-hand side of $\mathcal{L}_z$ and $\mathcal{L}_\phi$ are over thematic word, and simply we have $\mathcal{L}_z + \mathcal{L}_\phi = 0$. Since all the tokens are forced to have the same label, the classifier term is $\mathcal{L}_l = \sum_{t=1}^{T} \log p(l_t; h_t) = 0$. Also $\mathcal{L}_\theta = 0$, since it is the KL divergence between two equal distributions. Overall, $\mathcal{L}$ reduces to $\sum_{t=1}^{T} \log p(w_t; h_t)$, indicating that the model is just maximizing the log-model evidence based on the RNN output. $\square$

## B    Dataset Details

Table A1 provides details on the different datasets we use. Note that these are the same datasets as have been used by others [49].

## C    Implementation Details

For an RNN we used a single-layer LSTM with 600 units in the hidden layer, set the size of embedding to be 400, and had a fixed/maximum sequence length of 45. We also present experiments demonstrating the performance of basic RNN and GRU cells in Table 1a. Note that although our embedding size is higher, we are use an end-to-end training manner without using pre-trained word embeddings. However, as shown in Table 1a, we also examined the impact of using lower dimension embeddings and found our results to be fairly consistent. We found that using pretrained word embeddings such as word2vec could result in slight degradations in perplexity, and so we opted to learn the embeddings from scratch. We used a single Monte Carlo sample of $\theta$ per document and epoch. We used a dropout rate of 0.4. In all experiments, $\alpha$ is fixed to 0.5 based on the validation set metrics. We optimized our parameters using the Adam optimizer with initial learning rate $10^{-3}$ and early stopping (lack of validation performance improvement for three iterations). We implemented the VRTM with Tensorflow v1.13 and CUDA v8.0. Models were trained using a single Titan GPU. With a batch size of 200 documents, full training and evaluation runs typically took between 3 and 5 hours (depending on the number of topics).

## D    Perplexity Calculation

Following previous efforts we calculate perplexity across $D$ documents with $N$ tokens total as

$$\text{perplexity} = \exp\left( \frac{-\sum_{d=1}^{D} \log p(\mathbf{w}_d)}{N} \right), \tag{6}$$

| Dataset | Vocab | Training | | | Development | | | Testing | | |
|---|---|---|---|---|---|---|---|---|---|---|
| | | # Docs | # Sents | # Tokens | # Docs | # Sents | # Tokens | # Docs | # Sents | # Tokens |
| APNEWS | $7,788$ | $50K$ | $0.7M$ | $15M$ | $2K$ | $27.4K$ | $0.6M$ | $2K$ | $26.3K$ | $0.6M$ |
| IMDB | $8,734$ | $75K$ | $0.9M$ | $20M$ | $12.5K$ | $0.2M$ | $0.3M$ | $12.5K$ | $0.2M$ | $0.3M$ |
| BNC | $9,769$ | $15K$ | $0.8M$ | $18M$ | $1K$ | $44K$ | $1M$ | $1K$ | $52K$ | $1M$ |

Table A1: A summary of the datasets used in our experiments. We use the same datasets and splits as in previous work [49].

where $\log p(\mathbf{w}_d)$ factorizes according to eq. 1. We approximate the marginal probability of each token $p(w_t; \beta, h_t)$ within $d$ as

$$p(w_t; \beta, h_t) = \int p(\theta) \sum_{k=1}^{K} \sum_{l_t=0}^{1} p(w_t|z_t = k, l_t; h_t, \beta) p(z_t|l_t, \theta) p(l_t; h_t) \, d\theta.$$

$$\approx \frac{1}{S} \sum_{s=1}^{S} \sum_{k=1}^{K} \sum_{l_t=0}^{1} p(w_t|z_t = k, l_t; h_t, \beta) p(z_t = k|l_t, \theta^{(s)}) p(l_t; h_t)$$

$$= \frac{1}{S} \sum_{s=1}^{S} \sum_{k=1}^{K} \theta_k^{(s)} p(w_t|z_t = k; h_t, \beta) p(l_t = 1; h_t) + p(w_t; h_t) p(l_t = 0; h_t), \quad (7)$$

Each $\theta^{(s)} \sim q(\theta|w_{1:T}, \gamma)$ is sampled from the computed posterior approximation, and we draw $S$ samples per document.

# E  Text Generation

The overall generating document procedure is illustrated in Algorithm 1. We use <SEP> as a special symbol that we silently prepend to sentences during training.

---
**Algorithm 1** Generating Text
---
**Input:** sequence length ($l$)
**Output:** generated sentence
  $i \leftarrow 0$
  $w_0 \leftarrow$ <SEP>
  $\mathbf{w} \leftarrow [w_0]$
  **repeat**
    $i \leftarrow i + 1$
    $\theta^{(s)} \sim q(\theta|\mathbf{w})$
    $w_i \sim p(w_i|\mathbf{w})$
    $\mathbf{w} \leftarrow [\mathbf{w}, w_i]$
  **until** $i < l$
  **return w**
---

We limit the concatenation step ($\mathbf{w} \leftarrow [\mathbf{w}, w_i]$) to the previous 30 words.

# F  Generated Topics

See Table A2 for additional, randomly sampling topics from VRTM models learned on APNEWS, IMDB, and BNC (50 topics).

# G  Generated Sentences

In this part we provide some sample, generated output explain how we can generate text using VRTM. To this end, we begin with the start word and then we proceed to predict the next word given all the previous words. It is worth mentioning that for this task, the labels of stop and non-stop words are marginalized out and the model is predicting these labels best on RNN hidden states. This conditional probability is

$$p(w_t|w_{1:t-1}) = \int p(\theta) \sum_{k=1}^{K} \sum_{l_t=0}^{1} p(w_t|z_t, l_t; h_t, \beta)$$

$$p(z_t|l_t, \theta) p(l_t; h_t) \, d\theta. \quad (8)$$

Computing Eq. 8 exactly is intractable in our context. We apply Monte Carlo sampling $\theta^{(s)} \sim q(\theta|w_{1:T}, \alpha)$. It is too expensive to recompute $\theta$ with each word generated. To alleviate this problem,

| Dataset | #1 | #2 | #3 | #4 | #5 | #6 | #7 | #8 | #9 |
|---|---|---|---|---|---|---|---|---|---|
| **APNEWS** | dead | washington | soldiers | fund | police | state | car | republican | city |
| | killed | american | officers | million | death | voted | road | u.s. | residents |
| | hunting | california | army | bill | killed | voters | line | president | st. |
| | deaths | texas | weapons | finance | accusing | democrats | rail | campaign | downtown |
| | kill | residents | blaze | billion | trial | case | trail | candidates | visitors |
| **IMDB** | films | horror | pretty | friends | script | comedy | funny | hate | writing |
| | directed | murder | beautiful | series | line | starring | jim | cold | wrote |
| | story | strange | masterpiece | dvd | point | fun | amazed | sad | fan |
| | imdb | killing | intense | channel | describe | talking | naked | monster | question |
| | spoilers | crazy | feeling | shown | attention | talk | laughing | dawn | terribly |
| **BNC** | king | house | research | today | letter | system | married | financial | children |
| | london | st | published | ago | page | data | live | price | played |
| | northern | street | report | life | books | bit | love | poor | class |
| | conservative | town | reported | years | bible | runs | gentleman | thousands | 12 |
| | prince | club | title | earlier | writing | supply | dance | commission | age |

Table A2: Nine random topics extracted from a 50 topic VRTM learned on the APNEWS, IMDB and BNC corpora.

Figure A1: We examine the impact $\alpha$ has on the induced variational approximations $q$ from a 10 topic VRTM (selected due to dev perplexity performance). **Left:** effect of $\alpha$ on $\mathsf{H}(\theta)$ , **Right:** Appropriate choice of $\alpha$ also reduces $\mathsf{H}(\phi)$.

we can use a "sliding window" of previous words rather than the whole sequence to periodically resample $\theta$ [28, 8]. In this, we maintain a buffer of generated words and resample $\theta$ when the buffer is full (at which point we empty it and continue generating). We found that splitting each training document into blocks of 10 sentences generated more coherent sentence. Table 4 illustrates the quality of some generated sentences.

### G.1 The Effect of Hyperparameters on Topic Selectivity

Following previous work in language modeling that take the sparsity into consideration [1, 32], we sought for a selective token topic assignment with low entropy. First, we slightly overload the notation of entropy to define an average of non stop word-topics entropy and similarly document entropy: $\mathsf{H}(\phi) = -\frac{1}{T_n} \sum_{t:l_t=1} \sum_{k=1}^{K} \phi_t^k \log \phi_t^k$, and $\mathsf{H}(\theta) = -\frac{1}{T_d} \sum_{k=1}^{K} \theta^k \log \theta^k$, where $T_n$ and $T_d$ are the total number of non-stop words in the document and the total number of documents, respectively. Second, the Dirichlet distribution can easily be parameterized to generate sparse samples. Now we provide some intuitive explanation. In our setting, the equality $\mathcal{L}_z + \mathcal{L}_\phi = \frac{1}{S} \sum_{s=1}^{S} \sum_{t:l_t=1} \sum_{k=1}^{K} \phi_t^k \log \frac{\theta_k^{(s)}}{\phi_t^k}$ holds, which is the (negative) KL-divergence between $\phi$ and $\theta$ parameters. Moreover, on the $\mathcal{L}_\theta$ side, $\theta^{(s)}$ samples are controlled by the prior distribution. Overall, selectivity of $\phi$ strongly depends on the choice of prior parameters. As shown in Fig. A1, not only we can control the value of $\mathsf{H}(\theta)$, but also we can reduce $\mathsf{H}(\phi)$ by tuning the prior parameters without the need of any other regularizer.