[Reviews · NeurIPS 2020]

Review 1

Summary and Contributions: The paper proposes a new recurrent topic model, VRTM, which seems an extension of TopicRNN by not marginalizing the topic assignment. Even with such simple change, the proposed model outperforms the previous models in perplexity.

Strengths: In the proposed model, they include z_t, which is discrete topic representation in the model. When a discrete representation is involved in neural networks, the reparameterization trick is applied especially as in VAE. However this paper does not use it.

Weaknesses: In the paper, the equations could be more organized to see the contribution compared to the previous world.

Correctness: The proposed method looks correct. However, for the empirical results, the paper could have been better with more visualization (e.g., topic representation in 2D space).

Clarity: The paper could have been better if the equations are organized. I guess the paper has some issues for reproducibility in its current version.

Relation to Prior Work: They mentioned that their work is similar to TopicRNN and discussed the difference. As they said, the difference seems small, though the proposed method outperforms the previous one in a single experiment result (Table 1). However, it would be helpful to clarify the difference in section 3.

Reproducibility: No

Additional Feedback: The title includes "without reparameterization trick", but the paper does not mention much about it (maybe because they don't use it). However, the authors can explain more clearly how come we don't need reparameterization trick. As usual in deep learning, a better implementation could make some improvements. So, could we see some experiment results with several variations of VRTM, to check how much improvement we actually get by the proposed contribution? Some analysis could be helpful about Tables 3 and 4 compared to other methods. In line 198, the paper mentions "theorem" but there is no theorem. In line 150, the paper says "q(z_t=k | w_t, l_t; \phi_t) = 1/K". Should it be trainable?


Review 2

Summary and Contributions: In this paper, the authors attempted to utilize neural variational inference to construct a neural topic model with discrete random variables, and proposed one model, namely VRTM, which combine1. The exploration of combining RNNs and topic models is interesting and significant, which can help topic models to handle sequence text and capture more text information than the bag-of-word model, which is prevalently utilized in LDA-based topic models. 2. The proposed model, namely VRTM, can capture both the local, syntactically-driven lexical dependencies and global, thematic dependencies in text by modeling the thematic and non-thematic words with dynamic strategies. Specifically, when facing the thematic words, VRTM uses both the RNN and topic model predications to generative the next word; however, when facing the syntactic words, only the output of the RNN is utilized to predict the next word. In particular, during the generative process, the discrete topic assignment has been attached to each thematic word, which is beneficial for the Interpretability. 3. The authors have theoretically proved that VRTM can degenerate into RNN with one specific assumption, which not only demonstrate the flexibility of the proposed model, but also contribute to the understanding of proposed mechanisms for thematic vs. non-thematic words. s RNN models and topic models to capture both the local, syntactically-driven lexical dependencies and global, thematic dependencies in sequence of text. To be specific, the authors first designed one reasonable generative model, which can apply different strategies for generating thematic and syntactic words with different inputs, i.e., a mixture of LDA and RNN predications or just the output of the RNN. Then, in order to infer the latent variables in VRTM, one mean field-based neural variational inference approach has been developed without the reparameterization trick.

Strengths: 1. The exploration of combining RNNs and topic models is interesting and significant, which can help topic models to handle sequence text and capture more text information than the bag-of-word model, which is prevalently utilized in LDA-based topic models. 2. The proposed model, namely VRTM, can capture both the local, syntactically-driven lexical dependencies and global, thematic dependencies in text by modeling the thematic and non-thematic words with dynamic strategies. Specifically, when facing the thematic words, VRTM uses both the RNN and topic model predications to generative the next word; however, when facing the syntactic words, only the output of the RNN is utilized to predict the next word. In particular, during the generative process, the discrete topic assignment has been attached to each thematic word, which is beneficial for the Interpretability. 3. The authors have theoretically proved that VRTM can degenerate into RNN with one specific assumption, which not only demonstrate the flexibility of the proposed model, but also contribute to the understanding of proposed mechanisms for thematic vs. non-thematic words.

Weaknesses: 1. In the experiment section, the authors have designed a baseline that combines LDA and LSTM, namely LDA+LSTM. According to my understanding, this baseline can both capture the sequential information in text and provide the topic assignment for each word. I am curious to know the performance of this baseline in terms of the topic switch percent metric. 2. As the combination of RNNs and topic models, VRTM should also be evaluated with comparisons on the sentence generation capacity. It is better for the authors to demonstrate the performance of the sentence generation compared with some related models, such as RNNs, seq2seq model [1], and transformer [2], in terms of BLEU or ROUGE metrics. It is not sufficient to demonstrate the sentence generation capacity only by showing generated sentences. 3. In my opinion, it is necessary for the authors to analyze the impact of RNN-related component on the quality of learned topics and the effect of the topic model-related component on the quality of the generated sentences by comparisons, ablation experiments, or case studies, which would be valuable and provide more guidance for other researches in the related fields. [1] Sutskever, I., Vinyals, O., & Le, Q.V. (2014). Sequence to Sequence Learning with Neural Networks. ArXiv, abs/1409.3215. [2] Vaswani, A., Shazeer, N., Parmar, N., Uszkoreit, J., Jones, L., Gomez, A.N., Kaiser, L., & Polosukhin, I. (2017). Attention is All you Need. ArXiv, abs/1706.03762.

Correctness: 1. The technical claims and proof of the proposed model are correct. 2. There exist several weaknesses in the empirical methodology. It is better for author to add some additional experiments to demonstrate the performance of VRTM. (Detailed comments can be found in Weakness.) 3. Some other detailed comments: a) In line 126, should the word “thematic” be replaced by “non-thematic”? b) In line 216, the authors are supposed to add some citations at the end of the sentence “Following previous work”.

Clarity: Yes. The paper is well written.

Relation to Prior Work: Yes

Reproducibility: Yes

Additional Feedback:


Review 3

Summary and Contributions: Update: Thanks for clarifications on my reviews. I have updated my ratings based on your response to revised the paper accordingly. This paper presents variationally-learned recurrent neural topic model (VRTM) which consists of an RNN based language model to capture short-term syntactic/semantic dependencies and a generative topic model to capture long-term semantic dependencies. The proposed network explicitly models the topic for each word in a sequence during language modeling. In terms of contribution, it combines word-level generative processes of neural language model and probabilistic topic model using Dirichlet prior for the document-topic proportions. The proposed approach of maintaining word-level, discrete topic assignments and accounting for each word’s generating topic reports gains in generative tasks such as language modeling and improves topic quality. This work extends the composite language and topic modeling work by [7] with the two main differences: (i) taking into account the discrete topic assignments for each word via "topic-word" distributions without approximating them during training and inference. (ii) choosing the originally used (in LDA formulation) Dirichlet prior for "document-topic" distributions instead of the Gaussian prior.

Strengths: + This work is built upon [7] and introduces discrete modeling of word-level topics in the already existing joint topic and language modeling approach + incremental work, however reports gains in the language modeling and improving topic quality

Weaknesses: - incremental work and built upon [7]. The novelty/contribution is limited - missing motivation for introducing word-level discrete topics in extending the baseline [7]. An illustrative example is recommended. - problem is not clearly stated and formulated - The paper is build upon [7] however there have been several related works outperforming [7]. For instance, the most related works [44] employed neural topic model [NVDM, 25] instead of LDA. - The notion of using word-level topics is related to textTOvec (i) if the LDA model is replaced by an auto-regressive neural topic model (DocNADE) [19]. A quantitative evaluation and comparison with textTOvec is missing. - missing clear explanation of choosing Dirichlet distribution over Gaussian distribution - unclear use of terminology e.g. "without the Reparametrization trick" in title of the paper. Please clarify how is it related to [17]. - Figure 1(a) needs more details to explain the joint topic and language modeling architecture and connect (using symbols) the formulation discussed in the methodology section

Correctness: Beyond clarifications (mentioned in point 5.), the claims are supported by experimental evaluation. Empirical methodology: - missing details about the experimental setup (see point 7). - For a fair comparison, the authors should report scores of the rerun of (atleast) two baselines: (1) basic-LSTM, and (2) TopicRNN to verify the experimental setup and reproducibility. - In Table 1, the related works use 300 dimensional input embeddings initialized with Google word2vec embeddings, however this work employed 400 dimensional input embeddings. Therefore, the comparison with the prior works is not fair. - Perplexity formula in section D (Appendix) is incorrect: it does not account for the average over the number of documents in the test set? - In Table 4, topics used in generating sentences are missing (for example: top-5 topics in the average document-topic proportion) which are used for generating words based on the topic sample and RNN hidden units. - Line 79: Missing reference. textTOvec [(i)] introduces language structures in topic models. - missing details about the architecture and/or hyperparameters of the proposed model - Is pretraining of language model done before joint/combined training? - Architecture and hyperparameter settings of the MLP encoder, mentioned in section 3.2.1 line 177, is missing.

Clarity: - Section 3.2.1: in line 173 the elementwise multiplication should be between two tensors/vectors of same shape but c_v being a scalar (as described in line 170). Also, this multiplication will scale up the original embedding vector e_v by a factor of c_v^{TM} which do not simply correspond to the masking of non-thematic word vectors. Replacing with an indicator function would filter/mask non-thematic words. - Section 3.2.1: in line 177 no details about the MLP architecture to model "word-topic" distributions. - Notation for topic model vocabulary size changes from line 109 to lines 169-170. - Unclear notation. For instance, use bold-lowercase for vector and lowercase only for scalars. Additional comments: - line 60: "individual word assignments" -> "individual topic-word/word-topic assignments"? - line 93 unclear: how the generated topics are conditioned on the assigned topics? - line 126: "thematic word" -> "non-thematic word".

Relation to Prior Work: Discussed in 2 above.

Reproducibility: Yes

Additional Feedback: - clear notation scheme - a table of notations would boost reading - Transformer based models such as Transformer-XL and XLNet provides a way for language modeling at the document-level and capturing long-range syntactic and semantic dependencies. It would be interesting to see how the proposed model compares with such models. - The authors have provided an alternative to topic coherence evaluation but it would be interesting to see corresponding topic coherence evaluation scores and the correlation between topic-coherence and switchP. - In Table 1, it would be interesting to see the performance of the proposed model with LSTM/GRU cells in place of the vanilla-RNN cells as LSTM/GRU cells have been used in language modeling component in the most recent related works. Missing References: (i) textTOvec: DEEP CONTEXTUALIZED NEURAL AUTOREGRESSIVE TOPIC MODELS OF LANGUAGE WITH DISTRIBUTED COMPOSITIONAL PRIOR


Review 4

Summary and Contributions: This paper proposes a variational topic model which incorporates syntactic structure into the generative process, rather than working purely with word frequencies. In contrast to prior work, the authors explicitly instantiate the topic as part of the model, rather than integrating them out, instead parameterizing them through a distribution given by an autoencoder. They study the effects of this on the model's performance. UPDATE: after response and discussion, I continue to think this is a reasonable paper and am inclined to keep my score the same.

Strengths: This is a good paper. The authors' model is well-formulated, clearly stated, and its assumptions make sense. The idea to introduce autoencoders into topic models along with recurrent networks for modeling syntax has been studied before, but no dominant model has emerged and it is still unclear how to set this task up correctly, which is thus an important research question in topic models. In this paper, I found the divide of words into thematic and non-thematic words a particularly interesting modeling choice, and the short theorem which demonstrates that the model recovers a standard RNN-based language model if a uniform distribution is used over the topics is a nice touch to help with understanding. The experiments are comprehensive, with the authors comparing their proposal against many different methods on test set perplexity. Evaluating topic models well is a known difficult problem due to their unsupervised nature, so presenting a few of the generated topics is a nice touch (albeit a risky one - I have submitted model output in one of my own prior submissions because I thought it was good, only to have the paper rejected because the referees incorrectly accused me of cherry-picking, which years later I am still insulted by and upset about). Including model output is particularly helpful because it allows practitioners to subjectively gauge how well the model can be expected to perform, by simply reading the topics discovered on a standard corpora they are likely to have worked with before.

Weaknesses: The main downside is the use of a mean field assumption in the variational distribution of the topics. Mean field assumptions are known to perform poorly in many cases. Another issue is the use of a plain Bernoulli variable in the generative process to determine whether a word is thematically relevant, because in general this ought to depend on the word's context, such as other nearby tokens. These weaknesses could be explored as avenues of improvement in future work. The other downside is that all of the experiment datasets are tiny, with the largest having only 20m tokens, and the authors infer not very many topics. Developing scalable topic models is also an area of active research, and it is reasonable for the authors not to focus on this because the paper is on a different topic.

Correctness: Everything appears to be methodologically correct and I do not see any major issues. A good number of different competitor methods are benchmarked - a nice touch, especially since not papers on topic models release their source code, and good implementation can be challenging. Code is available.

Clarity: The paper is largely well-written. Below are some minor points of improvement. Some of the terminology is a bit ambiguous. For instance, I have never heard the term "neural variational inference" before - the authors say this is a synonym for variational autoencoders - the latter term is nowadays so standard that they should adopt it. Additionally, the authors follow the deep learning community and use "learning/inference" to refer to what statisticians call "inference/prediction". I suggest the terms "learning/prediction" because the topic modeling community is composed of a mix of machine learners and statisticians, as well as people in the social sciences and digital humanities, so avoiding the overloaded term "inference" seems helpful. Directly below eqn. 3, it is unclear what the authors mean by q(z_t) being defined similarly to p(z_t). I assume this is defined as the output of the autoencoder. It would be good to clarify this.

Relation to Prior Work: The authors do a nice job mentioning various prior work on neural topic models, which are still a relatively new and growing area, and their own different modeling choices are clearly written in Section 2 and Section 3.

Reproducibility: Yes

Additional Feedback: Please clarify the comment about eqn. 3. Please also consider adding a diagram to the paper, common in many deep learning papers, which shows the generative process with the encoder, decoder, and RNN, and in particular how different networks interact with each other in the loss. This would significantly improve clarify.

[Author Response · NeurIPS 2020]

We thank the reviewers for their in-depth reviews, and will use them to make the final version as clear as possible. Our
code, which was included in our supplementary material submission, will be made publicly available for reproducibility.
We will correct the missing notations and typos, we apologize for these oversights. Thank you for suggesting other
related work: we will expand our discussion with these.

Re: discussion/comparison: We have tried to provide both experimental and analytical discussion about word-level
topic modeling in conjunction with auto-regressive models without marginalizing out the topics. Our theorem shows
that under specific conditions our model reduces to a recurrent LM and our experimental results show that our topic
coherency outperforms variational LDA and Mallet LDA, covering both topic modeling and auto-regressive aspects.

We have reported perplexity per word consistent with prior work; we wil clarify the equation. We used a fixed max
length of 90 words per doc, with one layer and 200 hidden units in the recurrent structure. For all the experiments the
Dirichlet prior distribution parameter is 0.5. We will report these in the experiments section. As "RNN" can be used to
describe a number of different specific recurrent cells, we will add experimental results where we specifically compare
to a traditional RNN cell, an LSTM cell, and a GRU cell.

**[R1]**: We will reorganize the equations to make the analytic comparisons between our approach and previous work
more clear. Thank you for this suggestion. In our model definition, we are not sampling the discrete latent variable
from the posterior variational distribution and the discrete expectations are calculated in closed form. Without resorting
to the reparametrization trick, in our theorem (page 6 on top), we have shown that our model reduces to a simple RNN
just by assuming all the tokens are non-thematic words. We should note that our proof for this relies on the closed form
calculation and not sampling. We aim to design a language model that can preserve this word-level topic information.
The remarkable difference between our model and the TopicRNN paper is preservering this topic information. We have
imposed the simple uniform probability assumption just for the non-thematic words. This mechanism not only helps
the topic model distinguish between the thematic and non-thematic words, but also it leads to stable training, since in
the topic model part the gradients for the non-thematic words are zero and just the RNN part would be updated.

**[R2]**: We have included both variational LDA and Mallet LDA results for the switchP part to show that VRTM can also
be considered as a topic model in terms of topic coherency, we will add the results for LDA+LSTM in the final version.
However, we note that in LDA+LSTM an LDA model is trained and then used in a recurrent LM; this is in contrast to
our approach which allows both the topic & language model to be learned and updated jointly. We agree that examining
the generative capabilities of these types of recurrent models is important, but we believe that doing rigorously and
comprehensively needs its own study and is beyond the scope of this particular work. We provide output sentences as
examples so readers may make their own qualitative assessments on the strengths and limitations of our methods.

**[R3]**: We reconsidered core decisions made by TopicRNN, such as not marginalizing out topics and the doc prior. The
Dirichlet can *easily* be parameterized to generate sparse samples just by tuning the prior distribution parameters.

We will update our discussion to include textTOvec. We note that its topic modeling component $h^{DN}$ does not
marginalize over the topics in a traditional sense, as it passes topic parameters through an activation function, effectively
compressing the topic signal prior to any word generation. While not precisely the same as previous efforts, this is in
contrast to what we advocate, which is specifically conditioning a word's generation on a particular topic (and as topics
aren't observed, marginalizing after any activation function).

Re: implementation/results: Although we have employed 400 dimensional input embeddings, our word embeddings
are learned from scratch, which do not have the massive pretraining of other methods. Our early experiments showed
that our model and results were consistent even with other sizes like 300; we can include these numbers in the camera
ready. Our masked embedding is obtained by both masking the non-thematic words and multiplying the thematic word
embedding to their frequency. Since the topic models are mostly trained based on the frequency of (thematic) words our
aim is to define the embeddings in a way that (i) the topic model part neglects the non-thematic words, so the coefficient
for these words is zero. (ii) the thematic words with higher frequencies are emphasized. By adding this coefficient the
gradients flow will increase for the thematic words. We will clarify the notation. We have reported SwitchP rather than
other topic modeling metrics is that it makes the very intuitive yet simple assumption that "good" topics will exhibit a
type of inertia. We will update the paper with this motivation and add additional information to example output.

**[R4]**: We will add a diagram to clarify the encoding/decoding process. We view using the simpler mean field
approximation as a benefit and are excited to explore more expressive approximations in future work. Although we have
used the plain Bernoulli random variable in the generative process but in the joint probability definition the parameters
of this Bernoulli are learned in an auto-regressive manner where the information from the previous tokens play the role
to draw a thematic or non-thematic word. "Neural Variational Inference" is from [25]. We will clarify this point &
apologize for any confusion. Re: $q(z_t)$ and $p(z_t)$: For both $p$ and $q$ we have assumed the non-thematic words have the
uniform distribution. This construction motivates the topic model part to distinguish between thematic and non-thematic
words, and is used in the proof of the theorem.

[Meta-Review · NeurIPS 2020]

Reviews are all on the accept side: 1 top 50% of accepted and 3 marginally above threshold. Only R4 (strong accept) intervened in the discussion. As the main reason for calling this paper borderline was limited novelty compared to [7], I had to proceed to a detailed comparative rereading of this paper to [7]. In my opinion, this approach is very different from [7]. While the authors presented it as only introducing a small modeling difference from [7], this has a huge impact on everything, in particular the resulting DNN architecture and the inference process. But failure to appreciate the novelty may come from clarity issues. As pointed by R1, "the equations could be more organized to see the contribution compared to the previous world.", and I do not think one can fully understand and reproduce the model in its present description. We note that the authors also attached their code and stated in the rebuttal: "R1: We will reorganize the equations to make the analytic comparisons between our approach and previous work more clear", and "R4: We will add a diagram to clarify the encoding/decoding process." In summary: - I believe there is enough novelty in this paper to be accepted. - Absence of fully understandable descriptions seem to be an issue at every level of this paper: equations, model description, training procedure, experiments. Authors have stated they will clarify all of this, but will they have enough space? - 2 reviewers mentioned the confusing use of 'reparameterization' in the title: as the Authors have not addressed it in their rebuttal, let me bring this as a third reminder. Another reason for accept: language modeling approaches with explicit topic modeling put a much stronger burden of description on the authors than back boxes such as GPT-3 or T5, and authors should not be over-penalized for that effort.